# Alternatives in Education—Evaluation of Rat Simulators in Laboratory Animal Training Courses from Participants’ Perspective

**DOI:** 10.3390/ani11123462

**Published:** 2021-12-05

**Authors:** Melanie Humpenöder, Giuliano M. Corte, Marcel Pfützner, Mechthild Wiegard, Roswitha Merle, Katharina Hohlbaum, Nancy A. Erickson, Johanna Plendl, Christa Thöne-Reineke

**Affiliations:** 1Institute of Animal Welfare, Department of Veterinary Medicine, Animal Behavior and Laboratory Animal Science, Freie Universität Berlin, 14163 Berlin, Germany; Mechthild.Wiegard@fu-berlin.de (M.W.); Katharina.Hohlbaum@fu-berlin.de (K.H.); EricksonN@rki.de (N.A.E.); Thoene-Reineke.Christa@fu-berlin.de (C.T.-R.); 2Institute of Veterinary Anatomy, Department of Veterinary Medicine, Freie Universität Berlin, 14195 Berlin, Germany; Giuliano.Corte@fu-berlin.de (G.M.C.); office@myhumanx.com (M.P.); Johanna.Plendl@fu-berlin.de (J.P.); 3Institute for Veterinary Epidemiology and Biostatistics, Department of Veterinary Medicine, Freie Universität Berlin, 14163 Berlin, Germany; Roswitha.Merle@fu-berlin.de; 4MF 3—Experimental Animal Research and 3R—Method Development and Research Infrastructure, Robert Koch-Institute, 13353 Berlin, Germany

**Keywords:** 3R principle, humane education, training, alternative, laboratory animals, EU Directive, survey, SimulRATor, laboratory animals science courses, refinement

## Abstract

**Simple Summary:**

Training on live animals in laboratory animal science (LAS) courses is legally defined as an animal experiment. For stringent implementation of the 3R (replace, reduce, refine) principle, five rat simulators are currently available which provide training of handling and routine procedures. As these simulators seem to have great benefit for all users, the aim of this study is to investigate the simulators’ impact on the 3R principle from the course participants’ perspective, who can best evaluate their learning efficacy which, in turn, defines their 3R potential. Thus, the simulators were evaluated by 332 course participants of 27 specialized LAS courses by completing a practical training workshop and a paper-based two-part questionnaire, integrated in the official course schedule. The results revealed strong support for simulator-based training and it was considered a useful supplement in LAS training. However, the simulators currently available may not completely replace training on a live animal and improvements are necessary. As these results are also reflected in literature data on simulator training in other fields of education and training, more research regarding novel simulators and their development is needed, in order to ensure an even more comprehensive protection of laboratory animals in education and training in future.

**Abstract:**

In laboratory animal science (LAS) education and training, five simulators are available for exercises on handling and routine procedures on the rat, which is—beside mice—the most commonly used species in LAS. Since these simulators may have high potential in protecting laboratory rats, the aim of this study is to investigate the simulators’ impact on the 3R (replace, reduce, refine) principle in LAS education and training. Therefore, the simulators were evaluated by 332 course participants in 27 different LAS courses via a practical simulator training workshop and a paper-based two-part questionnaire—both integrated in the official LAS course schedule. The results showed a high positive resonance for simulator training and it was considered especially useful for the inexperienced. However, the current simulators may not completely replace exercises on live animals and improvements regarding more realistic simulators are demanded. In accordance with literature data on simulator-use also in other fields of education, more research on simulators and new developments are needed, particularly with the aim for a broad implementation in LAS education and training benefiting all 3Rs.

## 1. Introduction

In their pioneering work “The Principles of Humane Experimental Technique”, Russell and Burch stated “[…] it is widely recognized that the most humane treatment of experimental animals, far from being an obstacle, is actually a prerequisite for successful animal experiments” [1]. Their concept of the 3R principle which intends to replace animals in experiments, reduce the number of animals used and refine any suffering in the procedures whenever possible, has become a fundamental part of laboratory animal science (LAS) [2,3] and has been legally implemented for all laboratory animals in the European Union (EU) since the implementation of the European Directive 2010/63 (2010/63/EU) [4]. In order to reinforce the 3R principle in experiments and to refine procedures, European Member States have to ensure an adequate education and training for all personnel prior to their commencement of working with laboratory animals [4,5,6]. For this purpose, recommendations for education and training were established by experts for the European Commission [7,8] and, in many countries, personal licensees are qualified by species-specific laboratory animal training courses following these suggestions. Most of these courses focus on mice and rats [9,10] which are the most commonly used species for animal experiments [11,12] and include exercises on live animals, in order to provide the necessary manual skill acquisition required prior to conducting experiments. As the use of animals for educational purposes is itself legally classified as an animal experiment due to the potential pain, suffering, distress, or lasting harm imposed on the animals used [4], a 3R dilemma in LAS education and training emerged, as stated in our previous study [13]. While the use of live animals in education and training is still indispensable since non-animal alternatives cannot entirely fulfil the mandatory high standard of education and training, animals used for educational purposes require the same 3R principle and protection as do animals used for other experimental purposes. 

Although many alternatives have been established [14,15,16,17,18] to reduce or replace live mice and rats, e.g., in theoretical education [19,20,21,22,23,24,25,26,27] or practice of dissection [28,29,30,31,32] and suturing [33,34,35,36], according to our previous study, however, resources for practical training of handling and routine procedural techniques seem to be limited [13]. Beside toys, do-it-yourself-interventions [13,37] may also be applied. More advanced resources include rabbit silicon ears [38] and, perhaps the most advanced applications, five rat and one mouse simulator [39,40,41,42,43,44]. These simulators may potentially solve the dilemma of education and training regarding mice and rats as they aim to mimic the target species in size and anatomy and are designed to cover the most relevant techniques for LAS courses [13]. In spite of their unique market position as an alternative training resource for practical training on mice and rats, almost no data seems to be available on them to the authors’ knowledge. Literature research in this regard had covered books, reviews, and reports on alternative training resources [15,20,27,45,46,47,48] and one research article focusing on the general outcomes of LAS training courses, solely stating the use of one of the available simulators during the course [49]. However, our previous study was the only one that seems to have dealt with a focused analysis on the implementation and satisfaction with current simulators for LAS courses. This study, based on a survey among LAS course trainers and supervisors, revealed a rather poor simulator implementation and methodological assessment [13].

Besides trainers and supervisors, the largest group of simulator users is primarily inexperienced personnel, i.e., participants of specialized LAS training courses offered to acquire the essential knowledge and skills for experiments. This target group, beside the animals used, may also benefit most from simulator-based training and may be most suitable in assessing the degree of adequacy of preparation for practice on live animals provided by the currently available simulators [39,40,41,42,43,44]. This learning efficacy, in turn, highly determines the role of simulators as an alternative or supportive resource in education and training.

In order to investigate the impact of these simulators on the 3Rs particularly for the laboratory rat in LAS education and training, this study aims to evaluate the perspective of LAS course participants concerning learning efficacy and methodological satisfaction with available rat simulators [39,40,41,42,43] as well as to determine requirements for potential novel simulator designs. 

## 2. Materials and Methods

### 2.1. Survey Protocol

For practical evaluation in regularly conducted German- or English-language LAS courses covering the species of the laboratory rat, a voluntary simulator training workshop and a two-part paper-based questionnaire (see Appendix A) were developed for the course participants.

#### 2.1.1. Design and Pretest of the Survey

The workshop was designed as an additional practical task prior to practical training on live rats and was integrated in the regular course programme of specialized LAS courses. It was designed following a structural and systematic approach recommended by the National Competent Authorities for education and training [7]. After a short project introduction, techniques were explained and demonstrated using a rat simulator for handling and restraint and by video for tail vein injection [50]. Then, the participants were randomly distributed into small groups of up to six people and each group was assigned to one simulator station, at which the participants could practice all techniques trainable on the simulator type available at the respective station. One of the five different types of simulators was available per simulator station. The simulator stations were set up conforming to training stations in skills labs [51] and included not only instruments and disposable materials but also a booklet including step-by-step instructions for each technique trainable on the specific simulator type (see Appendix A). The exercises were supervised by the workshop instructor and the trainers and supervisors of the course.

The survey was primarily designed as a one-part questionnaire to be conducted after practical training. All questions were designed in consideration of the recommendations of the National Competent Authorities [7], the FELASA (Federation of European Laboratory Animal Science Associations, and the GV-SOLAS (Gesellschaft für Versuchstierkunde/Society of Laboratory Animal Science), and according to survey guidelines published by the GESIS (Leibniz Institute for the Social Sciences) [52,53,54,55,56], the largest European infrastructure institute for social sciences [57]. For better comparability, questions regarding the assessment of simulators and requirements for new developments, which had been included in a previous study pertaining to LAS course trainers and supervisors [13], were further processed and adapted to course participants. 

The draft of this one-part questionnaire underwent three consecutives cognitive pre-tests [58], each followed by a comprehensive revision. Finally, the survey concept—workshop and questionnaire—was pretested with 31 participants of a total of two LAS courses at the Freie University Berlin [59] in 2018. The workshop was easily implemented in the original course schedule. However, a two-part questionnaire seemed more adequate with the first part being conducted immediately after the simulator-workshop and the second part after the practical exercises on live rats. This setup was expected to produce more valid data, as many pre-test participants stated to prefer a methodological assessment of the simulator immediately after practice. Thus, the questions were arranged in two questionnaires, organizational instructions for a two-part questionnaire were added, and few questions were reworded upon suggestion by the pre-test participants. 

The final draft entailed a two-part mixed-typed questionnaire of 27 questions. For assessment questions, six-point Likert scale sets were used, with “1” being the best assessment or most appropriate and “6” being the worst assessment or least appropriate. Other question types included multiple-choice, numerical input, and open-ended questions. 

In the first part of the questionnaire, four questions pertained to methodological workshop and simulator training performance assessment, as well as material-related difficulties during practical simulator exercises and optional feedback messages. In the second part, the exercise on the live rat, the simulator’s anatomical correctness and its learning efficacy with regard to live animal training were to be assessed. Moreover, the participants were asked to specify methodological requirements for further developments, provide information on favorable or unfavorable aspects of the simulator used, and note suggestions for improvement. Demographic questions and optional feedback messages were included at the end of the second part.

This improved and more suitable two-part survey was evaluated in a third pre-test with 15 participants of a LAS course, which entailed only minor changes in wording. Furthermore, the English version was assessed by a native speaker for correctness and comprehensibility (for final version, see Appendix A).

#### 2.1.2. Practical Evaluation

The study design was reviewed and approved by the Ethics Committee of the Freie Universität Berlin (Ethic approval number: ZEA-Nr. 2021–012.). The study included a privacy policy which was provided to the participants prior to their strictly voluntary engagement in the study. Furthermore, all participants were offered the possibility to terminate their participation at any given time with no need for explanation.

In total, 13 simulators were used for evaluation, two of types A and E and three of types B, C, and D. In each course, all five simulator types were evaluated, except for the end of the study, at which all three copies of type C failed due to material defects (see Table 1). 

In order to ensure a maximum of six participants per station, multiple stations per simulator type were set up in larger courses. 

The practical evaluation was carried out between October 2018 and July 2019 in nine German- and 18 English-language LAS courses provided at overall 13 different institutions in Germany, Switzerland, the Netherlands, and Austria, which had volunteered to take part in this evaluation during the advertisement and conduction of a previous survey [13]. These courses granted the qualification to conduct animal experiments and were set up according to the EU Directive 2010/63 [4,5,6] or the framework for education and training published by the European Commission [7].

For comparability, the evaluation was always carried out by one of the two project members—in English or German—according to the language of the LAS course. Furthermore, part one of the questionnaire was conducted immediately after the simulator workshop, part two after the practical training on live rats. In summary, 347 course participants—158 in German and 189 in English courses—took part in the anonymous and voluntary survey.

### 2.2. Statistical Analysis

Survey data were transferred in Microsoft Excel 2018. Data of short-text open-ended questions were transmitted via numerical or letter code in order to categorize the responses for further analysis. The entire data set was checked twice for completeness, transcription errors, or typos prior to analysis. Finally, 332 out of 347 responses were used for statistical analysis, as 14 respondents had skipped more than 25% of the questions either in part 1 or part 2 and one response showed several signals of response bias [60,61,62,63]—straight-line and primacy effects—in the response pattern. For this study, exclusively, questions concerning the methodological assessment of the rat simulators and an assessment of their learning efficacy prior to practice on live rats, as well as regarding requirements for further developments were analyzed.

The data set was analyzed descriptively via IBM SPSS Statistics 26. For Six-point-Likert-scale items and numerical input, data were reported as median values. For multiple-choice questions and coded short-text open-ended questions, data were reported as absolute response frequencies. Data of uncoded long-text open-ended question were analyzed individually.

The descriptive analysis was first carried out for the total data set of 332 responses and then separately for responses according to the simulator type evaluated, the course language, the 13 different course providers and for those respondents stating “no”, “a little bit”, or “a lot of” previous experience in handling rats, each followed by a comprehensive check against deviations and outliers. Potential correlations were proofed via Mann-Whitney-U-Tests and Kruskal-Wallis-Tests for assessment questions and data derived from multiple-choice and coded short-text open-ended questions were tested via Pearson correlation. The results were illustrated in tables or figures created by Microsoft Excel 2018 and Microsoft Word 2018.

## 3. Results

### 3.1. Demographics

Total of 197 participants stated to be female, 118 male, 3 diverse, and 14 abstained. The median age of the respondents was calculated to be 26.5 years and most respondents stated student (*n* = 133) or academic employee (*n* = 105) status. Moreover, 27 technical assistants and 19 apprentices were among the participants. “Other positions” were indicated 36 times which mainly included animal care takers as well as doctoral students and doctoral researchers. Twelve were abstained. Among the participants, biology and human or veterinary medicine with the subdisciplines of biochemistry, biomedicine or neuroscience were expressed as most common background disciplines. Overall, 72 participants expected to work with rats in the future and 55 expected to work with rats and mice.

Regarding previous experience in handling rats, 240 participants reported “no”, 61 participants “a little bit”, and 19 participants “a lot of” previous experience, whereas twelve did not answer this question. About 17 participants reported that they had used other simulators before, e.g., simulators for cows, dogs or humans, and three participants also indicated previous training experience with a rat simulator. 

Total of 65 participants evaluated rat simulator A (participants group “A”), 78 participants rat simulator B (participants group “B”), 53 participants rat simulator C (participants group “C”), 73 participants rat simulator D (participants group “D”), and 63 participants rat simulator E (participants group “E”) (see Tables 1–4).

### 3.2. Participants’ Evaluation of the Simulator-Training Workshop (Questionnaire Part 1)

The results showed no significant differences among the participants with respect to the simulator type used. All participants indicated that the number of participants per simulator station, which ranged from one to six, was appropriate to them, as well as the training duration on the simulator, which was approximately 20 min per participant (see Appendix A questions used for analysis part 1, question 2). Furthermore, overwhelmingly positive feedback was provided concerning the workshop itself and also a great endorsement for the previous simulator training (citations from questionnaire part 1, message box): 


*“For i.v. and blood injection the rat [simulator] was very helpful. It´s already difficult to inject the needle in the vein. Now I know, to what to pay attention when I use a real rat.”*
(feedback from a participant without experience in handling rats who used rat simulator type A) 


*“The model was really helpful, but some improvements can be done”*
(feedback from a participant with a little bit of experience in handling rats who used rat simulator type C)


*“I was afraid of rats before and could not handle rats correctly. Now, I’m becoming more optimistic on how to do it. Thank you very much.”*
(feedback from a participant with a little bit of experience in handling rats who used rat simulator type E).

### 3.3. Participants’ Evaluation of the Practical Simulator Training

#### 3.3.1. Participants’ Methodological Assessment of the Practical Training with Simulators

Regarding the total data set including all simulator types (see Table 2, column 2), training of handling, restraint, and administration via the tail vein as well as blood sampling from the tail vein, which could be practiced on all simulator types, were predominantly rated as “quite good” (median grade of 2.00) in the first part of the questionnaire. Apart from this, however, training of restraint via “scruffing” or via “under the shoulder grip” was rated rather poorly with a median grade of 3.00 (“slightly good”).

Regarding the separate analysis for each simulator type (see Table 2, columns 3–7), the participants who had practiced on the simulator types C (*n* = 51–52 for each of the four restraint techniques), D (*n* = 71–73), and E (*n* = 62–63) assessed the restraint techniques—especially “scruffing”—more poorly than those who had practiced on the types A (*n* = 64) and B (*n* = 77–78). Furthermore, simulator type D received the most inferior rating for “scruffing” with a median grade of 5.00 by the participants (*n* = 71).

Training of oral gavage, which was practiced by a total of 190 participants on the simulator types A (*n* = 60), B (*n* = 73), and E (*n* = 57), was rated “slightly good” in summary for all three simulator types (see Table 2, line 8, column 2). In comparison between the individual simulator types, type A received an outstanding rating of “quite good” (median grade of 2.00) (see Table 2, line 8, columns 3–7). 

With respect to the techniques only to be practiced on a specific simulator type, participants who practiced on simulator type C (*n* = 42/45) rated blood sampling from the saphenous vein and cardiac puncture very poorly with a median grade of 5.00 and 6.00, respectively. However, participants who practiced on simulator type D (*n* = 62) assessed the exercise ear punch as “quite good” with a median grade of 2.00. Additionally, simulator type E also received “quite good” or “slightly good” (median grade of 2.00 or 3.00, respectively) evaluations for subcutaneous and intramuscular injection training, respectively (*n* = 53/49/58). 

In summary and comparing each simulator type, simulator type A consistently received “quite good” ratings (median grades of 2.00) by the participants (see Table 2).

#### 3.3.2. Participant Statements on Material-Related Difficulties

In total, oral gavage (*n* = 69), restraint via scuffing (*n* = 65), and blood sampling from the tail vein (*n* = 62) were the techniques most frequently reported in which material-related difficulties occurred, of which oral gavage was solely trainable on simulator types A (*n* = 60), B (*n* = 73), and E (*n* = 57). The two other techniques could be practiced on all five simulator types.

Comparing the results for each simulator type, “scruffing” was mentioned particularly and most frequently in terms of technical difficulties by participants who had practiced on simulator type D—in almost half of all cases (*n* = 33).

Oral gavage, which was trainable solely on simulator types A, B, and E, was the most frequently mentioned technique for simulator types B (*n* = 33) and E (*n* = 23) featuring material-based difficulties. 

Furthermore, regarding the techniques that could only be practiced on simulator type C, cardiac puncture (*n* = 23) followed by blood sampling from the saphenous vein (*n* = 15) was the most frequently reported technique. Besides, very few participants (*n* = 7) stated material-related difficulties during training of intramuscular injection on simulator type E. 

In terms of absolute data, participants who had practiced on simulator type A mentioned material-related difficulties significantly less often than participants who had practiced on simulator types B-E (see Table 2, line 21). 

### 3.4. Participants’ Evaluation of the Practical Training with Live Rats

#### 3.4.1. Participants’ Personal Assessment of Their Practical Training Performance on Live Rats

The participants evaluated their own performance on live rats in the second part of the questionnaire distributed after the practical training on live animals. Except for the techniques of blood withdrawal from the orbital sinus and the lateral tail veins, which were evaluated as “slightly good” (median grade of 3.00), the performance of all the other techniques was assessed to be “quite good” (median grade of 2.00) by all participants (see Table 3, column 2).

Comparing the participants’ self-assessments (see Table 3, columns 3–7, lines 1–20), the worst evaluation was provided for blood sampling from the orbital sinus with a median grade of 4.00 by the participants who had practiced on simulator type D (*n* = 23) (see Table 3, column 6, line 17).

Concerning handling and restraint, the participants who had previously practiced on simulator types C (*n* = 48), D (*n* = 68), and E (*n* = 56) assessed themselves more poorly in “scruffing” with a median grade of 3.00 (type C) and 2.50 (type D and E) as compared to groups of types A and B with a median grade of 2.00. In addition, participants who had practiced on simulator type C (*n* = 41) evaluated their performance of the “under the shoulder grip” more poorly with a median grade of 3.00 as compared to all other participants who had trained on types A, B, D, and E. 

With regard to training of administration via the lateral tail vein, participants who had previously practiced on simulator types C (*n* = 30), A (*n* = 47), and D (*n* = 49) assessed themselves more poorly with a median grade of 2.50 (type C) and 3.00 (type A and D) than participants of B and E with a median grade of 2.00.

For blood sampling from the tail vein, participants who had practiced on simulator types B (*n* = 61) and C (*n* = 38) revealed a more positive performance assessment with a median grade of 2.00 than the participants who practiced on types A (*n* = 47), D (*n* = 52), and E (*n* = 52) with a median grade of 3.00.

In performing oral gavage on the live rat—trainable solely on simulator types A (*n* = 40), B (*n* = 35), and E (*n* = 31)—exclusively the participants of simulator type D (*n* = 37) rated their performance with a median grade of 3.00 worse than all other participants with a median grade of 2.00—which also included participants of type C (*n* = 37) on which oral gavage was also not trainable.

Similarly, only participants who had practiced on simulator type E (*n* = 29) rated their performance of blood sampling from the saphenous vein with a median grade of 3.00 more poorly than the participants who had practiced on types A–D (median grade of 2.00). The best self-assessment was given by the participants who had practiced on simulator type E (*n* = 19) for their performance of ear punch with a median grade of 1.00, although this technique was solely trainable on simulator type D (see Table 3, lines 2–20 columns 3–7,).

#### 3.4.2. Participants´ Statements on the Most Demanding Techniques Performed on Live Rats

In the second part of the questionnaire, the participants were asked to state the three techniques on the live rat which, in their opinion, were particularly demanding for the performer. Blood sampling from the tail vein (*n* = 110) and heart (*n* = 99) as well as oral gavage (*n* = 86) were mentioned most frequently overall (see Table 3, line 21, column 2). Additionally, “scruffing” was mentioned in fourth position (data not shown).

Comparing the data regarding the simulator types on which the participants had previously practiced on, only very minor differences were evident (see Table 3, line 21, columns 3–7). Interestingly, participants who had practiced on simulator type C—the only simulator type on which cardiac puncture was trainable—mentioned this technique in fourth instead of first to third position in comparison to all other participants (see Table 3, line 21, column 5). Of note, for participants who had practiced on simulator type E, oral gavage was the fourth, i.e., not among the first three most frequently reported techniques (data not shown). Instead, “scruffing” (see Table 3, line 21, column 7) was mentioned. 

### 3.5. Participants’ Assessment of the Learning Efficacy after the Practical Training on Rats

In the second part of the questionnaire, the participants also provided a personal evaluation of the learning efficacy for the type of simulator which they had practiced on (see Table 4).

The data, summarized for all five simulator types, revealed that the participants consistently evaluated the learning efficacy for practicing handling, restraint, and tail vein administration as well as blood sampling, techniques which could be practiced on all simulator types, as “slightly good” with a median grade of 3.00 (see Table 4, column 2). 

Comparing the results for each simulator type separately, participants who had practiced on simulator type C (*n* = 52/52) evaluated the learning efficacy in practicing handling and restraint via “scruffing” more poorly with a median grade of 4.00 (=“slightly bad”) for both techniques and participants of type D (*n* = 67) evaluated latter technique with a median grade of 5.00 (=“quite bad”) more poorly than the other participant groups A, B, and E with a median grade of 3.00 (=“slightly good”) which also pertained to group D for handling and the other restraint grips (see Table 4, lines 2–6, columns 3–7).

Regarding the techniques performed on the rat simulators’ tails, participants who had practiced administration (*n* = 56) and blood sampling via the tail vein (*n* = 59) on simulator type A evaluated the learning efficacy for these exercises with a median grade of 2.50 and 2.00, respectively, better than the other participant groups B–E with median grades of 3.00 each (see Table 4, lines 15, 19, columns 3–7).

Training of oral gavage, which could only be practiced on simulator types A (*n* = 50), B (*n* = 50), and E (*n* = 37), was attributed to an overall learning efficacy of a median grade of 3.00 (see Table 4, line 8, column 2) for all three types. When evaluated separately, the participants who had practiced on simulator type E (*n* = 37) assessed the learning efficacy more poorly with a median grade of 4.00 than the participants of A and B with a median grade of 3.00 (see Table 4, line 8, columns 3–7).

With respect to the techniques which could be only practiced on one simulator type, the learning efficacy for ear punch on simulator type D (*n* = 42) was rated as “quite good“ with a median grade of 2.00 (see Table 4, line 7, column 6). Cardiac puncture and saphenous vein blood withdrawal, trainable on simulator type C, were assessed as particularly poor by the participants (*n* = 36/32) with a median grade of 4.00 (see Table 4, lines 18, 20, column 5). Subcutaneous and intramuscular administration techniques, trainable solely on simulator type E, received a median grade of 3.00 by the participants (*n* = 48/41/35) (see Table 4, lines 10–12, column 7).

### 3.6. Methodological Requirements for a Novel Rat Simulator

When all participants were asked to select the five most relevant techniques which should be trainable on a new rat simulator, they most frequently selected blood sampling via the lateral tail vein, restraint, oral gavage, administration in the lateral tail vein, and cardiac puncture (see Figure 1). Data displayed no response differences between the participant groups regarding the different simulator types used or prior experience. 

### 3.7. Participants Comments and Feedback Messages after Practical Training with Live Rats

#### 3.7.1. Participants’ Comments on the Simulator They Practiced On

Finally, the participants were requested to indicate particularly favorable or unfavorable aspects of the simulator type they had practiced on and which improvements they suggested in three open-ended questions (questions used for analysis see Appendix A, questionnaire part 2, questions 11, 12, 13). The overall analysis revealed only limited differences in the responses regarding the simulator types.

Generally, the participants particularly favored training on the tail vein, especially the administration. Furthermore, according to the course participants’ opinion, initial personal barriers, fears, and doubts prior to practice on the live rat were diminished, especially for the inexperienced. In addition, practicing restraint and procedures step-by-step with no time-constraints was well appreciated. Regarding the individual simulator types, participants who had practiced on rat simulator type B particularly welcomed the viewing window for oral gavage and those who had practiced on type E favored training restraint.

However, participants strongly suggested an enhanced realism of simulators. These should imitate the rat’s proportions correctly and also include anatomical structures, e.g., in the tail or in the body to provide improved flexibility of the core and extremities and, hence, an increased transferability to live rats. Besides, overall haptics and skin consistency were to be improved. With respect to the different types of simulators, the participants who practiced on simulator type C mainly requested improvements concerning blood sampling from the heart and the saphenous vein whilst participants of type E requested improvements concerning oral gavage (citations from questionnaire part 2, questions 11, 12, 13).


*“You lose the initial shy before touching a rat; you can practice the procedures step by step and slowly to get sort of a routine”*
(positive comment from a participant without experience in handling rats who used rat simulator type B)


*“The material and its stiffness (everything quite hard)-more flexible would be more realistic”*
(negative comment from a participant without experience in handling rats who used rat simulator type C)


*“Improve the way the body feels and make sure, that you can actually grab the skin/fur behind the neck and back”*
(comment for improvements from a participant without experience in handling rats who used rat simulator type D)

#### 3.7.2. Participants Feedback Messages after Practical Training on Live Rats

In a final feedback (question used for analysis, see Appendix A, questionnaire part 2, feedback box), the participants strongly advocated simulator-based training as a supportive training method in LAS courses which, from their point of view, potentially reduces personal barriers or doubts prior to practice on live rats but, at the same time, indicated that exercises on existing simulator types cannot in any case replace practice on live rats-(citations from questionnaire part 2, feedback box):


*“Interesting, but some improvements are necessary”; “make the simulator more realistic”*
(feedback messages from two participants with a lot of experience in handling rats who used rat simulator type C)


*“Simulator increased confidence but reality of handling seems hard to simulate”*
(feedback message from a participant without experience in handling rats who used rat simulator type E)


*“I liked the training provided by the simulator. I believe, it is an important step prior working with the live animals. Specially because one gets to train the handling and fixation of animals, as well as, how to collect blood samples and injections”*
(feedback message from a participant with a little bit of experience in handling rats who used rat simulator type E).


*“Good option, but they have to be improved to be realistic (touch, rigidity). Vein injection is easy because we can see the “tube” that is simulating the vessel. In real rats sometimes that is not happening”*
(feedback message from a participant with a little bit of experience in handling rats who used rat simulator type D).

### 3.8. Impact of Experience in Handling Rats, Language, or Course Providers

The comparison of the median values, standard deviations as well as the Pearson correlation, Mann-Whitney-U-Tests and Kruskal-Wallis-Tests did not show any significant differences due to the different level of experience. In data analysis, only sporadic correlations in very few sub-questions for the course providers and language were found which showed neither an association between each other, nor a recognizable pattern.

## 4. Discussion

The clear advocation of simulator-based training among the participants in this survey which had also been expressed in a previous survey on course trainers’ and supervisors’ perspectives [13] implies the application of current simulators as competent refinement and reduction method in LAS training. This may be particularly useful for the inexperienced, as previous training on the simulators may decrease initial barriers or fears prior to training on the live animal as stated by the participants of this study numerous times. Both surveys, however, also demonstrate that the rat simulators currently available require improvement in order to be used in a broader 3R concept or as replacement. According to the responses provided in this and the previous survey [13], new developments of rat simulators should be optimized especially toward a more realistic design and at least provide the techniques of handling, restraint, oral gavage, and administration as well as blood sampling via the lateral tail vein.

### 4.1. Demographics and Study Design

#### 4.1.1. Demographics

The aim of the survey was to investigate the current simulators’ impact in LAS education and training. LAS personnel is required to receive specialized training prior to experimental engagement, which is, to our experience, mostly acquired via LAS courses. We assume that we were able to reach a representative number of study attendees with a total of 332 LAS course participants in 27 German- and English-language LAS courses of 13 different international providers, including universities and specialized service providers. Anyhow, the representative sample size cannot be calculated, which poses a limiting factor due to the lack of robust data on course numbers or on the providers in the respective countries, as stated previously [13,64].

Our estimation is, however, underlined by the background diversity of the attendees of this survey which reflects all relevant employment profiles and disciplines commonly present in LAS, i.e., trainees, undergraduates, PhD students, postdocs or senior scientists belonging to various fields of life science, such as nutritional science, biology and pharmacy or medicine, and more particularly, molecular, dental, veterinary, and human medicine. 

#### 4.1.2. Study Design

By integrating the practical simulator workshop and the questionnaire-based evaluation into regular LAS courses, we not only evaluated the most common preparatory method regarding animal experiments, but we were also able to evaluate the simulators under realistic conditions. Due to defined time slots in the courses, each participant could practice solely on one of the five simulator types available during the simulator workshop. Hence, the participants were prepared in different types of techniques prior to the training on live rats. Although this concept bears the risk of a minor bias between the five simulator groups, we chose this study design since our aim was to investigate the individual impact and learning efficacy of each of the five different simulator types A–E separately. If participants had been able to practice on several simulator types, the effect of the repetition and the series of the simulators used would have had to be accounted for, not only in the statistical analysis but also in the entire study concept. Furthermore, the duration of the workshop and the questionnaire would have increased significantly if every participant were to practice on every simulator type for equal time periods. Hence, the integration into the regular LAS courses may have been far more complicated if not impossible. Likewise, such extended training sessions prior to exercise on live animals does not seem feasible within the recommended course time tables [65]. 

This practical evaluation in official LAS courses, of course, also bears a minor potential for bias regarding the different course providers and courses visited during the survey period. However, we consider the results received from different course providers and courses to be comparable to each other: 

First, one aim in protecting laboratory animals according to the 3R principle is to provide an equivalent education and training for everyone performing animal experiments regardless of the course and country [7,66,67]. Therefore, all courses evaluated here follow the recommendations of the EU Commission [7] and courses with a certification of FELASA [67] or GV-SOLAS [68] additionally comply with the regulations of these particular organizations. Furthermore, all courses met their respective national legal regulations [69,70,71,72,73,74,75] which conform to the EU Directive 2010/63 [4]. Since all courses of our study either took place in a Member State of the European Union or Switzerland, the legal provisions of which also conform with the requirements of the EU Directive [4], we further assume comparability of these courses. Nevertheless, a certain variance between individual courses can never be excluded which, however, may lead to greater robustness of the data and reflect reality more closely. In addition, this heterogeneity also poses a challenging factor for researchers and developers of rat simulators aiming at bringing new developments to a large market for alternatives in LAS training.

Potential variance was considered throughout the study and minimized by appropriate measures. Regarding the simulator training workshop and the evaluation, we estimate a strong comparability between the courses, as the workshops were conducted in the same fashion in English or German language by one of a total of two different workshop instructors of the project team. In each course, five, and after the loss of simulator type C due to technical defects, four different simulator types were evaluated. Furthermore, the workshop was always integrated as a separate training session prior to exercise on the live rat and the questionnaires were always carried out directly after the simulator workshop for part one and after the practical training on the live rat for part two in each course.

The total number of respondents per simulator type varied between 53 and 78. We assume that these variations barely had any effect on the validity of the study’s results, since a sufficient number of respondents took part by which the statistical analysis could be performed for all objectives. 

When the participants were asked to randomly distribute themselves among the simulator stations in the workshop, our main focus was a numerically even distribution, aiming to provide an effective training with a maximum of four to six participants per working station. This concept leads to minute numeric variations between the simulator groups. 

By the use of the simulator types in duplicate or triplicate, we could use the simulator types more efficiently and evaluate courses taking place simultaneously or courses with a very large number of participants–which consequently increased the overall representativeness of our results.

Since the aim of this study exclusively pertained to the analysis of the simulators regarding methodological assessment, learning efficacy, and novel design requirements, and not to the impact of course providers or language, the minute sporadic correlations in rare sub-questions for these were not taken into further account.

### 4.2. Participants‘ Evaluation of the Simulator Workshop

The participants evaluated the workshop organization overall as “good” in the median. Thus, it may be assumed that the respective simulator types which were used and evaluated by the participants did not pose an influence regarding the opinion on the workshop in general. 

The given number of participants per simulator station and training duration per simulators were confirmed to be appropriate according to the participants’ responses. Therefore, with respect to the time available in regular LAS course schedules, we suggest the integration of a separate simulator training with one to six participants per simulator station and a training duration of about 20 min per participant as an effective preparation for live animal work. This set-up seems feasible in providing each participant the possibility of practicing the techniques at his or her own pace as many times as necessary until he or she feels confident enough to continue with training on animals. Since time is usually tight in these courses, the workshop duration may be shortened by increasing the number of simulators available at each station or in the course. 

The participants’ overall positive feedback concerning the exercises on the simulators coincides with that provided by LAS course trainers and supervisors in a separate online survey as published previously [13]. A similar assessment is reflected in related disciplines, such as education and training in veterinary medicine [76,77,78,79,80,81,82,83,84,85,86,87,88,89], human medicine [90,91,92,93], biology or pharmacy, and other life sciences [94], which already apply simulators in a more established fashion. 

In training veterinary students, most likely the best comparison to LAS training due to the key importance of the animal´s behavior for performance, numerous studies have assessed the positive effect of simulator-based training prior or in direct comparison to exercises on live animals with regard to students of different training levels [77,78,79,80,81,82,87,88,95,96]. Nevertheless, studies also demonstrate the limitations of current simulator-based training [78,82,96,97] which, in turn, underscore the necessity of live animal training also in veterinary medicine with regard to the mandatory ”Day One Competences” and the legal framework provided by the European Association of Establishments for Veterinary Education (EAEVE) [98,99,100]. 

Moreover, analysis of course evaluations [101] and surveys conducted with participants of LAS courses [102] concerning their attitude toward LAS courses, i.e., the use of live animals for training or the evaluation of particular course aspects revealed that the respondents considered practice of handling and routine procedures, including practice on a live animal, as a very important part in preparatory training. Therefore, simulator-based training may be considered a meaningful supplement in the sense of the 3R principle for LAS training which, however, should be further developed in future. 

### 4.3. Participants‘ Evaluation of the Training on the Simulators

Concerning simulator type B, course trainers and supervisors attributed methodological dissatisfaction for oral gavage and restraint [13] as compared to the positive evaluations of the training and learning efficacy of all trainable techniques including oral gavage and restraint by the course participants. Since the other simulator types were not assessed in the previous survey due to their poor course implementation, comparisons for these were not possible.

The findings of this study on course participants may be subjected to further analysis in future projects, e.g., by means of implementing a control group. This was omitted in our survey from the onset, since it would have imposed an entirely different questionnaire concept as most of the questions provided here would not have been relevant to the control group, i.e., the assessment of simulator’s practice or learning efficacy. Moreover, it would have hampered the workshop organization or execution and most likely also the LAS course itself. Course providers are particularly committed to offer a comparable course schedule and training opportunity to each participant during the LAS course which, hence, would have disadvantaged participants of the control group.

Furthermore, future projects may include animal-based welfare parameters. These were not assessed here due to the focus on training and learning efficacy from the trainees’ perspective. Moreover, the assessment of animal welfare parameters would have necessitated an amendment of each of the course providers’ animal experimental license at their local authorities [69,70] which was simply not feasible. Hence, future studies on animal welfare parameters may be performed as single-center study. 

The course participants’ assessments of simulator training, learning efficacy, and performance on the live rat revealed that simulator-based training may be considered a feasible training resource in LAS courses, with room for improvements. The participants rated their performance on the live rat mainly as “quite good”, regardless of whether the technique had previously been practiced on a simulator and on which type. However, the simulator training exercises were consistently rated positively—i.e., as “quite good”—while the learning efficacy was consistently rated slightly less positively—i.e., as “slightly good”—for nearly all techniques. These findings coincide with the overall positive evaluation of simulator training on the one hand and the suggestions for improvement reflected in the open-ended remarks section, implying a certain degree of improvability on the other. 

Concerning the different types of simulators, minor differences were evident. Types A and B may be more appropriate for practical training of handling and restraint, for which particularly types C and D appeared to be poorly suited, also coinciding with the responses given for material-based difficulties during practice. Taking into account the minor differences between the simulator types and considering that the five types of simulators provide a different range of techniques, the choice of type or of several different types of simulators to be used in LAS courses must be carefully considered so that the training benefits from the advantages of each individual simulator type.

Improvement for techniques already trainable on simulators should be particularly considered for blood sampling from tail vein, restraint, oral gavage, and also blood sampling from the heart since these techniques were considered most demanding to perform on the live rat and displayed most material-based difficulties. Furthermore, they were regarded as most relevant for new developments by the course participants. 

### 4.4. Participants‘ Requirements for a Novel Simulator

Based on the results of this survey and in comparison with our previous survey on course trainers and supervisors [13], a new rat simulator should at least provide preparatory training for handling, restraint, oral gavage, and administration, as well as for blood sampling via the lateral tail vein. These techniques were among the most frequently requested by the course participants and the course trainers and supervisors [13]. Moreover, handling and restraint–particularly via “scruffing” or via “over the shoulder grip”, were mostly practiced by the course participants in this study and were, together with oral gavage, one of the most commonly practiced techniques on conscious rats in regular LAS courses from trainers’ and supervisors’ point of view [13].

Based on our results, a novel simulator should also include cardiac puncture and at least one other common injection technique for rats as an option. Cardiac puncture was the fifth most requested technique for a new rat simulator by the course participants and, coinciding, the previous survey considered it one of the most commonly practiced techniques on anesthetized rats [13]. This technique is highly demanding, particularly for the inexperienced, since it does not offer visual control and requires numerous manual skills being executed simultaneously [103]. Besides, it is also emotionally demanding to puncture the heart as central organ. Despite a state of deep anesthesia, gasping or other reflexes may occur which must be considered humane endpoints and lead to termination of the exercise. Therefore, it is not surprising that the course participants stated this technique as one of the most demanding on live rats. Moreover, cardiac puncture seems to be one of the subjects for improvability in future simulator developments considering the poor assessments of training and learning efficacy for simulator type C, the sole simulator offering practice of this technique. Thus, we estimate that a rat simulator which provides cardiac puncture for practice will in this regard strengthen the 3Rs in LAS education and training. 

Furthermore, we recommend implementing at least one other common injection technique [104,105,106,107,108], subcutaneous (s.c.), or intraperitoneal (i.p.) injection. I.p. and s.c. injection were rated in 7th and 8th position by the course participants while i.p. injection was one of the ten most requested techniques by the trainers and supervisors [13]. Besides, i.p. injection was one of the most commonly practiced techniques by the course participants and both techniques, i.p. and s.c. injection, were two of the most commonly practiced techniques in regular LAS courses from the trainers’ and supervisors’ perspective [13]. Despite literature reports on several disadvantages of i.p. administration in rats [109,110,111,112,113,114], it is yet one of the most commonly performed techniques according to our results in this study and the previous study [13]. The choice of this technique needs to be carefully considered with regard to the advantages, disadvantages, and risks of the respective procedure, also with regard to the imperative of choosing the least burdening route of application possible, which should be covered in the theoretical part of the training course. Although the technique itself may be debatable, practicing this technique is particularly important due to its shortcomings while it is still commonly being performed.

The current FELASA recommendations [115], in accordance with the Directive 2010/63/EU [4,5,6] and the related European Commission guidance paper [7], suggest live animal practice—on conscious animals for handling and immobilization and on anesthetized animals for minimally invasive procedures such as applications or blood sampling—after preparatory training on alternatives in LAS training courses, which demonstrate an effective implementation of the 3Rs. Preparatory training of these techniques on simulators strictly follows these recommendations, as it can provide training in using needles, syringes, and other material under roughly realistic conditions which also gives the participant the opportunity to practice the technique until he or she has successfully mastered the manual skills before proceeding to the exercise on live animals.

Importantly, training on live animals—either conscious or anaesthetized, according to the FELASA recommendations, the Directive 2010/63/EU and the related European Commission guidance paper, as stated above—should involve only those persons who will definitely apply the respective techniques in the sense of the indispensability of this animal experiment for training purposes and, hence, also with regard to the 3R principle.

As the training of handling, restraint, and oral gavage on a conscious rat with its voluntary reactions and reflexes crucial to training itself cannot be mimicked by a non-animal alternative to date, a rat simulator providing for these techniques may offer a great potential for the 3Rs, especially in the sense of refinement. By extensive pre-training of course participants, their improved security and accuracy, may significantly reduce the animals’ stress and thus, also aid in adequate preparation for later animal experiments, in which learning curves are to be avoided as strictly as possible as these may distort experimental data. However, the use of simulators should not only be restricted to LAS training for initial certification but also for maintenance of skills later on.

Hence, a novel simulator should not only be designed to cover most of the routinely relevant techniques in terms of a broad application of the 3Rs but also consider animal welfare-related aspects for each technique separately by e.g., preferentially offering techniques which are commonly performed on conscious animals but may also include critical or highly demanding techniques despite these being performed under general anesthesia.

Additionally, as the participants especially liked verification options, e.g., a viewing window as provided by simulator type B, realistic options for performance control should also be discussed for novel simulators which allow trainees to recognize their mistakes, so that they can learn from them.

Under these conditions, a new simulator development will strengthen the application of the 3R principle in LAS courses and should be combined with other important measures in the sense of the 3R principle, such as practical exercise on cadavers and prior handling of animals or positive reinforcement in order to provide the animals with the possibility of becoming accustomed to handling and/or certain procedures—even if non-invasive—and, hence, to reduce stress.

## 5. Conclusions

From course participants’ point of view, this study demonstrates that simulator-based training is a suitable addition to LAS courses and hereby may contribute to an even more extended implementation of the 3Rs and animal welfare in education and training. It may also benefit experimental studies conducted later. However, the currently available simulators do not entirely meet the requirements of LAS course participants and bear means of improvement. These need to be assessed in detail with regard to a broader application of the 3Rs while also taking animal welfare-related aspects for each technique separately into account. In order to optimize and strengthen the use of simulator-based training, further research on simulator implementation and animal-welfare-related parameters as well as novel developments are needed, with a special focus on realistic transferability.

## Figures and Tables

**Figure 1 animals-11-03462-f001:**
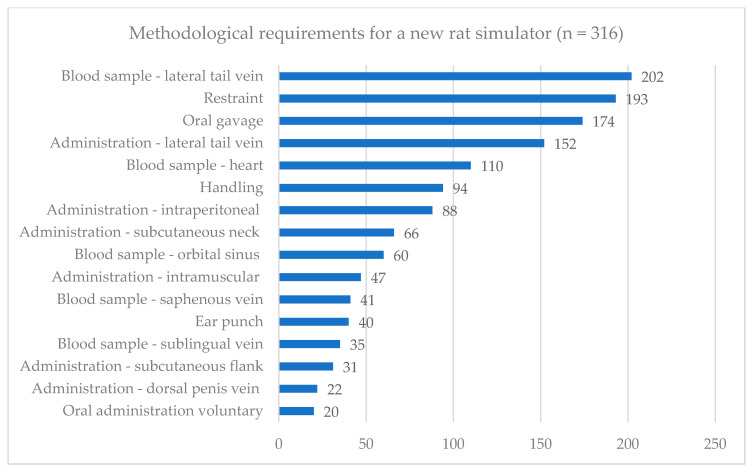
Descriptive analysis of methodological requirements for a novel rat simulator. Distribution of replies regarding the trainable techniques most relevant for a novel rat simulator development derived from the multiple-choice question “Please select from the list below the five techniques on the live rat, for which you consider a preparatory training on simulators to be particularly useful” (question used for analysis see Appendix A, questionnaire part 2, question 4). Absolute number of responses is shown for each technique comprising all respondents (*n* = 316).

**Table 1 animals-11-03462-t001:** Overview of the rat simulators evaluated by course participants in specialized LAS courses. Product names were anonymized.

Product Information	Evaluated Rat Simulators				
	Rat Simulator A	Rat Simulator B	Rat Simulator C	Rat Simulator D	Rat Simulator E
External appearance	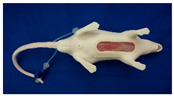	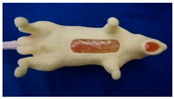	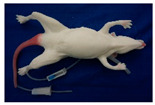	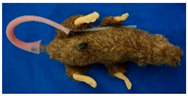	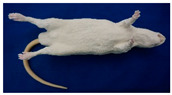
NORINA database record number [16]	5e236	88cf4	05ebd	457b1	f7a0d
Numbers of simulators per type used for practical evaluation	2	3	3 ^i^	3	2
Techniques practiced per simulator type	Handling	Handling	Handling	Handling	Handling
	Restraint 4 techniques	Restraint 4 techniques	Restraint 4 techniques	Restraint 4 techniques	Restraint 4 techniques
	Administration by oral gavage intravenous via tail vein	Administration by oral gavage intravenous via tail vein	Administration intravenous via tail vein	Administration intravenous via tail vein	Administration by oral gavage intravenous via tail vein subcutaneous (neck/flank) intramuscular
	Blood sampling via tail vein	Blood sampling via tail vein	Blood sampling via tail vein saphenous vein cardiac blood sampling	Blood sampling via tail vein	Blood sampling via tail vein
				Ear punch	
Instruments and materials provided for training at the simulator stations	Curved stainless steel feeding needle for rats 2 × 1 mL syringe (filled with water) ^ii^ 2 × 27 G plastic canula ^ii^ 1 × 1 mL syringe (empty) ^ii^ Paper swabs ^ii^ Booklet	Curved stainless steel feeding needle for rats 2 × 1 mL syringe (filled with water) ^ii^ 2 × 27 G plastic cannula ^ii^ 1 × 1 mL syringe (empty) ^ii^ Paper swabs ^ii^ Booklet	1 × 1 mL syringe (filled with water) ^ii^ 1 × Lancet length 3 mm ^iii^ 1 × 20 G plastic cannula ^iv^ 2 × 27 G plastic cannula ^ii^ 1 × 1 mL syringe (empty) ^ii^ Paper swabs ^ii^ Booklet	Scissor Style Ear Punch ^v^ 1 mL syringe (filled with water) ^ii^ 1 × 1 mL syringe (empty) ^ii^ 2 × 27 G plastic cannula ^ii^ Paper swabs ^ii^ Booklet	Curved stainless steel feeding needle for rats 5 × 1 mL syringe (empty) ^ii^ 1 × 1 mL syringe (filled with water) ^ii, vi^ 3 × 26 G plastic cannula ^ii^ 2 × 27 G plastic cannula ^ii^ Paper swabs ^ii^ Booklet
Specifications according to manufacturer’s manual	Application of water and performance control for oral gavage	Application of water and performance control for oral gavage	-	Fur, flexible head	Application solely of air allowed for oral gavage, subcutaneous and intramuscular administration, fluids allowed for tail vein

^i^ Not evaluated after 2019/05/10 due to defects. ^ii^ Disposable material was used according to the manufacture’s manual. Disposable material was provided by the course providers. ^iii^ Lancets (3 mm) were used instead of 28 G Lancets as stated in the manufacture’s manual. Lancets were provided by the course providers. ^iv^ 20 G plastic cannulas were used instead of 25 G as stated in the manufacture’s manual and were provided by the course providers. ^v^ Scissor Style Ear Punch was provided by the course providers. ^vi^ 1 mL syringes were used instead of 2 ml syringes as stated in the manufacture’s manual and were provided by the course providers.

**Table 2 animals-11-03462-t002:** Descriptive analysis of the participants´ methodological assessment of practical simulator training and material-related difficulties. Results are presented for all survey respondents (ALL) and separately for each simulator type (A–E). Absolute numbers of total respondents are presented in the headline. Lines 2–20 display the calculated median values, standard derivations, and absolute numbers of responses to the question “How well were you able to apply the procedural techniques on the simulator?” Responses pertained to 14 individual techniques assessed by a six-point Likert scale with “1 = extremely good, 2 = quite good, 3 = slightly good, 4 = slightly bad, 5 = quite bad, 6 = extremely bad” (question used for analysis, see Appendix A, part 1, question 1). Line 21 represents the three most frequently indicated techniques including the absolute number of responses to the open field question “If there were material-related difficulties using the simulator, please give us a brief description of these” (question used for analysis, see Appendix A, questionnaire part 1, question 3).

	Group of Participants	ALL	A	B	C	D	E
	(N)	N = 332	N = 65	N = 78	N = 53	N = 73	N = 63
1	Handling and routine procedures	Median value (Standard Deviation σ) Absolute number of responses (*n*)					
2	Handling (h)	2.00 (±1.42) *n* = 303	2.00 (±0.89) *n* = 60	2.00 (±0.91) *n* = 69	2.00 (±1.05) *n* = 50	2.00 (±1.17) *n* = 69	2.00 (±1.01) *n* = 55
3	Restraint–scruffing (r1)	3.00 (±1.58) *n* = 326	2.00 (±0.97) *n* = 64	2.00 (±1.02) *n* = 77	4.00 (±1.82) *n* = 51	5.00 (±1.47) *n* = 71	3.00 (±1.15) *n* = 63
4	Restraint–over the shoulder grip (r2)	2.00 (±1.07) *n* = 329	2.00 (±0.94) *n* = 64	2.00 (±1.16) *n* = 78	2.00 (±0.96) *n* = 52	2.00 (±1.17) *n* = 72	2.00 (±1.00) *n* = 63
5	Restraint–middle shoulder grip (r3)	2.00 (±1.09) *n* = 328	2.00 (±1.11) *n* = 64	2.00 (±1.20) *n* = 77	2.00 (±0.90) *n* = 52	3.00 (±1.28) *n* = 72	3.00 (±0.80) *n* = 63
6	Restraint–under the shoulder grip (r4)	3.00 (±1.19) *n* = 327	2.00 (±1.18) *n* = 64	2.00 (±1.23) *n* = 77	3.00 (±1.03) *n* = 51	3.00 (±1.37) *n* =73	2.50 (±0.91) *n* = 62
7	Ear punch (ep)	2.00 (±1.23) *n* = 62	-	-	-	2.00 (±1.23) *n* = 62	-
8	Oral gavage (og)	3.00 (±1.41) *n* = 190	2.00 (±1.03) *n* = 60	3.00 (±1.29) *n* = 73	-	-	3.00 (±1.69) *n* = 57
9	Oral administration voluntary (oa-v)	-	-	-	-	-	-
10	Administration–subcutaneous neck (sc-n)	2.00 (±1.28) *n* = 53	-	-	-	-	2.00 (±1.28) *n* = 53
11	Administration–subcutaneous-flank (sc-f)	3.00 (±1.34) *n* = 49	-	-	-	-	3.00 (±1.34) *n* = 49
12	Administration–intramuscular (im)	3.00 (±1.25) *n* = 58	-	-	-	-	3.00 (±1.25) *n* = 58
13	Administration–intraperitoneal (ip)	-	-	-	-	-	-
14	Administration–dorsal penis vein (iv-dp)	-	-	-	-	-	-
15	Administration–lateral tail vein (iv-tv)	2.00 (±1.22) *n* = 323	2.00 (±1.07) *n* = 65	2.00 (±1.28) *n* = 78	2.00 (±1.40) *n* = 49	2.00 (±0.90) *n* = 72	2.00 (±1.39) *n* = 59
16	Blood sample–sublingual vein (bs-slv)	-	-	-	-	-	-
17	Blood sample–orbital sinus (bs-os)	-	-	-	-	-	-
18	Blood sample–saphenous vein (bs-sv)	5.00 (±1.76) *n* = 42	-	-	5.00 (±1.76) *n* = 42	-	-
19	Blood sample–lateral tail veins (bs-tv)	2.00 (±1.26) *n* = 321	2.00 (±1.11) *n* = 64	3.00 (±1.42) *n* =75	2.00 (±1.29) *n* = 50	2.00 (±0.96) *n* = 72	2.00 (±1.32) *n* = 60
20	Blood sample–heart (bs-h)	6.00 (±1.58) *n* = 45	-	-	6.00 (±1.58) *n* = 45	-	-
21	Absolute number of responses for material-based difficulties	og (*n* = 69) r1 (*n* = 65) bs-tv (*n* = 62)	bs-tv (*n* = 14) iv-tv (*n* = 8) og (*n* = 8)	og (*n* = 33) bs-tv (*n* = 27) iv-tv (*n* = 13)	bs-h (*n* = 23) bs-sv (*n* = 15) r1 (*n* = 14)	r1 (*n* = 33) bs-tv (*n* = 11) iv-tv (*n* = 7)	og (*n* = 23) r1 (*n* = 15) im (*n* = 7)

**Table 3 animals-11-03462-t003:** Descriptive analysis of the participants´ performance assessment of practical training on live rats and the most demanding techniques. Results are presented for all survey respondents (ALL) and separately for each simulator type (A–E). Absolute numbers of total respondents are presented in the headline. Lines 2–20 display the calculated median values, standard derivations, and absolute numbers of responses to the questions “How well were you able to manage handling, restraint, and ear punching on the live rat? How well were you able to manage the following procedural techniques on the live rat?” Responses pertained to 19 individual techniques assessed by a six-point Likert scale with “1 = extremely good, 2 = quite good, 3 = slightly good, 4 = slightly bad, 5 = quite bad, 6= extremely bad” (questions used for analysis questionnaire part 2, question 1 and 2). Line 21 represents the three most demanding techniques including the absolute number of responses to the open field question “Which three techniques on the live rat are in your opinion particularly demanding for the performer?” (question used for analysis, see Appendix A, questionnaire part 2, question 3).

	Group of Participants	ALL	A	B	C	D	E
	(N)	N = 332	N = 65	N = 78	N = 53	N = 73	N = 63
1	Handling and routine procedures	Median value (Standard Deviation σ) Absolute number of responses (*n*)					
2	Handling (h)	2.00 (±0.96) *n* = 318	2.00 (±1.13) *n* = 64	2.00 (±0.84) *n* = 72	2.00 (±0.98) *n* = 51	2.00 (±0.91) *n* = 71	2.00 (±0.97) *n* = 60
3	Restraint–scruffing (r1)	2.00 (±1.21) *n* = 301	2.00 (±1.10) *n* = 59	2.00 (±1.17) *n* = 70	3.00 (±1.26) *n* = 48	2.50 (±1.37) *n* = 68	2.50 (±1.14) *n* = 56
4	Restraint–over the shoulder grip (r2)	2.00 (±1.08) *n* = 288	2.00 (±1.15) *n* = 55	2.00 (±0.97) *n* = 67	2.00 (±1.12) *n* = 47	2.00 (±1.11) *n* = 64	2.00 (±1.12) *n* = 55
5	Restraint–middle shoulder grip (r3)	2.00 (±1.07) *n* = 253	2.00 (±1.15) *n* = 53	2.00 (±0.96) *n* = 60	2.00 (±1.04) *n* = 41	2.00 (±1.06) *n* = 55	2.00 (±1.19) *n* = 44
6	Restraint–under the shoulder grip (r4)	2.00 (±1.10) *n* = 251	2.00 (±1.14) *n* = 52	2.00 (±1.05) *n* = 60	3.00(±1.05) *n* = 41	2.00 (±1.10) *n* = 54	2.00 (±1.20) *n* = 44
7	Ear punch (ep)	2.00 (±1.08) *n* = 102	2.00 (±1.38) *n* = 24	2.00 (±1.12) *n* = 23	2.00(±0.91) *n* = 16	2.00 (±0.79) *n* = 20	1.00 (±0.70) *n* = 19
8	Oral application voluntary (ov)	2.00 (±1.22) *n* = 61	2.50 (±0.89) *n* = 8	2.00 (±1.02) *n* = 16	2.00 (±1.50) *n* = 9	2.00 (±1.46) *n* = 18	1.50 (±0.82) *n* = 10
9	Oral gavage (og)	2.00 (±1.08) *n* = 161	2.00 (±1.13) *n* = 40	2.00 (±0.93) *n* = 35	2.00(±1.10) *n* = 18	3.00 (±0.93) *n* = 37	2.00 (±1.35) *n* = 31
10	Administration–subcutaneous neck (sc-n)	2.00 (±0.90) *n* = 225	2.00 (±0.84) *n* = 40	2.00 (±0.78) *n* = 51	2.00 (±1.07) *n* = 37	2.00 (±0.97) *n* = 51	2.00 (±0.85) *n* = 46
11	Administration–subcutaneous flank (sc-f)	2.00 (±0.90) *n* = 193	2.00 (±0.88) *n* = 33	2.00 (±0.89) *n* = 40	2.00 (±0.98) *n* = 32	2.00 (±0.93) *n* = 47	2.00 (±0.87) *n* = 41
12	Administration–intramuscular (im)	2.00 (±0.91) *n* = 153	2.00 (±0.79) *n* = 31	2.00 (±0.84) *n* = 35	2.00 (±1.28) *n* = 18	2.00 (±1.00) *n* = 36	2.00 (±0.75) *n* = 33
13	Administration–intraperitoneal (ip)	2.00 (±0.83) *n* = 276	2.00 (±0.82) *n* = 59	2.00 (±0.80) *n* = 60	2.00 (±0.82) *n* = 41	2.00 (±0.89) *n* = 62	2.00 (±0.81) *n* = 54
14	Administration–dorsal penis vein (iv-dp)	2.00 (±2.04) *n* = 42	3.00 (±0.49) *n* = 7	2.00 (±3.72) *n* = 10	2.00 (±1.27) *n* = 7	2.00 (±1.25) *n* = 13	2.00 (±1.14) *n* = 5
15	Administration–lateral tail veins (iv-tv)	2.00 (±1.29) *n* = 215	3.00 (±1.37) *n* = 47	2.00 (±1.17) *n* = 50	2.50 (±1.41) *n* = 30	3.00 (±1.31) *n* = 49	2.00 (±1.19) *n* = 39
16	Blood sample–sublingual vein (bs-slv)	2.00 (±1.08) *n* = 82	2.50 (±1.03) *n* = 16	2.00 (±0.89) *n* = 20	2.00 (±1.29) *n* = 11	2.50 (±1.31) *n* = 20	3.00 (±0.96) *n* = 15
17	Blood sample–orbital sinus (bs -os)	3.00 (±1.37) *n* = 94	3.00 (±1.26) *n* = 17	3.00 (±1.22) *n* = 20	3.00 (±1.44) *n* = 15	4.00 (±1.68) *n* = 23	3.00 (±1.00) *n* = 19
18	Blood sample–saphenous vein (bs-sv)	2.00 (±1.22) *n* = 135	2.00 (±0.89) *n* = 30	2.00 (±1.39) *n* = 26	2.00 (±1.21) *n* = 21	2.00 (±1.35) *n* = 29	3.00 (±1.27) *n* = 29
19	Blood sample–lateral tail veins (bs-tv)	3.00 (±1.30) *n* = 250	3.00 (±1.27) *n* = 47	2.00 (±1.39) *n* = 61	2.00 (±1.39) *n* = 38	3.00 (±1.28) *n* = 52	3.00 (±1.17) *n* = 52
20	Blood sample–heart (bs-h)	2.00 (±1.18) *n* = 213	2.00 (±1.30) *n* = 46	2.00 (±1.30) *n* = 46	2.00 (±1.17) *n* = 31	2.00 (±1.24) *n* = 47	2.00 (±0.88) *n* = 43
21	Absolute number of responses for most demanding techniques on live rats	bs-tv = 110 bs-h = 99 og = 86	bs-tv = 24 bs-h = 22 og = 19	bs–tv = 27 bs-h = 26 og = 20	bs-tv = 16 og = 13 bs-sv = 13 bs-h = 11	bs-h = 22 og = 21 bs-tv = 19	bs-tv = 23 r1 = 18 bs-h = 17

**Table 4 animals-11-03462-t004:** Descriptive analysis of the participants’ learning efficacy assessment of the simulator type which they had practiced on in the workshop. Results are presented for all survey respondents (ALL) and separately for each simulator type (A–E). Absolute numbers of total respondents are presented in the headline. Lines 2–20 display the calculated median values, standard derivations, and absolute numbers of responses to the question “How well did the simulator prepare you for the following techniques in the course training on the live rat?” Responses pertained to 14 individual techniques assessed by a six-point Likert scale with “1 = extremely good, 2 = quite good, 3 = slightly good, 4 = slightly bad, 5 = quite bad, 6= extremely bad” (question used for analysis questionnaire part 2, question 10).

	Group of Participants	ALL	A	B	C	D	E
	(N)	N = 332	N = 65	N = 78	N = 53	N = 73	N = 63
1	Handling und routine procedures	Median value (Standard Deviation σ) Absolute number of responses (*n*)					
2	Handling (h)	3.00 (±1.30) *n* = 308	3.00 (±1.15) *n* = 61	3.00 (±1.12) *n* = 70	4.00 (±1.49) *n* = 52	3.00 (±1.34) *n* = 67	3.00 (±1.37) *n* = 58
3	Restraint–scruffing (r1)	3.00 (±1.38) *n* = 312	3.00 (±1.06) *n* = 62	3.00 (±1.07) *n* = 74	4.00 (±1.66) *n* = 52	5.00 (±1.44) *n* = 67	3.00 (±1.35) *n* = 57
4	Restraint–over the shoulder grip (r2)	3.00 (±1.20) *n* = 299	3.00 (±0.98) *n* = 59	3.00 (±1.02) *n* = 70	3.00 (±1.34) *n* = 49	3.00 (±1.35) *n* = 65	3.00 (±1.20) *n* = 56
5	Restraint–middle shoulder grip (r3)	3.00 (±1.19) *n* = 270	3.00 (±1.04) *n* = 57	3.00 (±1.07) *n* = 62	3.00 (±1.22) *n* = 44	3.00 (±1.39) *n* = 60	3.00 (±1.12) *n* = 47
6	Restraint–under the shoulder grip (r4)	3.00 (±1.22) *n* = 269	3.00 (±1.08) *n* = 56	3.00 (±1.08) *n* = 63	3.00 (±1.40) *n* = 44	3.00 (±1.37) *n* = 60	3.00 (±1.13) *n* = 46
7	Ear punch (ep)	2.00 (±1.24) *n* = 42	-	-	-	2.00 (±1.24) *n* = 42	-
8	Oral application voluntary (ov)	-	-	-	-	-	-
9	Oral gavage (og)	3.00 (±1.40) *n* = 137	3.00 (±0.98) *n* = 50	3.00 (±1.38) *n* = 50	-	-	4.00 (±1.75) *n* = 37
10	Administration–subcutaneous neck (sc-n)	3.00 (±1.35) *n* = 48	-	-	-	-	3.00 (±1.35) *n* = 48
11	Administration–subcutaneous flank (sc-f)	3.00 (±1.26) *n* = 41	-	-	-	-	3.00 (±1.26) *n* = 41
12	Administration–intramuscular (im)	3.00 (±1.40) *n* = 35	-	-	-	-	3.00 (±1.40) *n* = 35
13	Administration–intraperitoneal (ip)	-	-	-	-	-	-
14	Administration–dorsal penis vein (iv- dp)	-	-	-	-	-	-
15	Administration–lateral tail veins (iv-tv)	3.00 (±1.21) *n* = 275	2.50 (±1.00) *n* = 56	3.00 (±1.10) *n* = 69	3.00 (±1.50) *n* = 41	3.00 (±1.18) *n* = 63	3.00 (±1.37) *n* = 46
15	Blood sample–sublingual vein (bs-slv)	-	-	-	-	-	-
17	Blood sample–orbital sinus (bs-os)	-	-	-	-	-	-
18	Blood sample–saphenous vein (bs-sv)	4.00 (±1.62) *n* = 32	-	-	4.00 (±1.62) *n* = 32	-	-
19	Blood sample–lateral tail veins (bs-tv)	3.00 (±1.25) *n* = 284	2.00 (±1.01) *n* = 59	3.00 (±1.24) *n* = 68	3.00 (±1.51) *n* = 44	3.00 (±1.16) *n* = 62	3.00 (±1.39) *n* = 51
20	Blood sample–heart (bs-h)	4.00 (±1.59) *n* = 36	-	-	4.00 (±1.59) *n* = 36	-	-

## Data Availability

The data presented in this study are available on request from the corresponding author. The data are not publicly available due to unanalyzed data not related to the subject of this study.

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
