# Peer review of "Alternatives in Education—Evaluation of Rat Simulators in Laboratory Animal Training Courses from Participants’ Perspective"

_animals, 2021, doi:10.3390/ani11123462_

Round 1

Reviewer 1 Report

This is an extremely well-designed and well-written study which deserves to be widely read by all those involved in the education and training of laboratory animal care staff and users. I congratulate the authors on the thoroughness of this project. The manuscript's English is also excellent.

The main aim is to investigate the impact of five rat simulators on participants on specialised courses in Laboratory Animal Science, to determine the learning efficacy of these simulators, and therefore their potential to implement the 3Rs of Russell & Burch (Replacement, Reduction and Refinement of animal expriments). 

Although there are published studies describing the efficacy of other simulators (e.g. computer simulations), and of simulators in human and veterinary medicine, much of our knowledge about the efficacy of mannikins in the context described in this paper (basic education and training in Laboratory Animal Science) is based on anecdotal evidence. Many mannikins have varying degrees of fidelity and discrimination, which can introduce bias into the way in which their value is reported by the participants. It is therefore of great interest to conduct detailed investigations such as the present study. The topic addressed in this paper is therefore highly relevant.

The authors were limited by the constraints of the courses, so it would have been difficult to improve the methodology. Actually, any such improvements would remove one of the strengths of this paper: evaluation under course conditions.

The conclusions are consistent with the evidence and arguments
presented and the authors address the main question posed.

The reference list is long and comprehensive, appropriate to the topic. Several of the co-authors are well-known, senior members of the Laboratory Science community, with a long track record of working in this area. This is reflected in the reference list.

Some additional comments for tables and figures:

Table 1: I don't see why footnote 5 is mentioned before footnote 4.

The word "by" should be added to footnote 5: "by the course providers".

The colour photos in this Table are a great help to the reader.

Table 2 doesn't seem to exist...I think the tables are numbered incorrectly.

Figure 1, and the students' comments under section 3.7, will be of particular interest to future students and course providers.

Reviewer 2 Report

In this article, the authors discuss the use of rat simulators in regulatory LAS training. The study presents, from the point of view of the participants, the impact of such a use in the learning of common gestures subsequently carried out in live rodents. This work follows a previous study presenting the impact of non-living surrogates in training from the point of view of trainers and supervisors.

In the context of increased animal protection and increased demands for the 3Rs, this study is very relevant. The context and the problematic are well introduced, the methodology used is rigorous and well described, from the design of the questionnaire to the data collection and analysis methods. The results are well presented and put into perspective. In the end, this is an impressive piece of work, rich in lessons and therefore worthy of wide dissemination.

The methodology is based on a questionnaire, previously tested, and adjusted, which was offered to many volunteer participants during formal training sessions. Importantly, in its final form, the questionnaire is in two stages, with questions asked immediately at the end of training on simulators and other questions whose answers were collected after training on live animals. This makes the study particularly interesting as the time between the two is short. It would probably have been more difficult to obtain a high response rate if this delay had been several weeks or months. Another key advantage is that this methodology was proposed to several official training courses meeting the same specifications and regulatory requirements and in several languages. The results show that the influence of language is weak, which makes the results of the study easily transposable. In addition, it should be noted that the questionnaire included items to be filled in as a score, with a panel of six answers which can be easily compiled, but also open-ended questions which are more difficult to analyse.

With regards to the representativeness of the total sample of participants and establishments, the authors' argument is perfectly valid. The methodological rigour that gives this large-scale study a good degree of robustness must be stressed once again. The participants responding to the questionnaires are perfectly representative of the staff encountered in regulatory training, notably in terms of their level of previous experience with animals.

The study highlights the value of simulators as a prerequisite to training on live animals. The first advantage is to remove any apprehension linked to the handling of a live animal; especially in naive participants. A second advantage is the flexibility of time for handling and learning. If the average handling time for each participant is about 20 minutes, each participant can perform the gesture at his/her own pace. A third advantage of this type of simulator, which is not emphasised here, is the possibility to repeat the exercise as many times as necessary, until the performance of the gesture is considered suitable. From this point of view, it appears that the five simulators tested are equivalent and useful.

The five simulators, tested simultaneously during the same sessions, allow the learning of different gestures or techniques, with some differences. I found the authors rather reserved about some of the results, which clearly show the shortcomings of certain models. The aim of the study was not to compare them, but the results of the survey clearly show the limitations of some simulators. This observation suggests that the choice of a particular simulator for a LAS training course must therefore be carefully thought through, and that considering not one but maybe two or more simulators seems relevant to compensate for their respective limitations.

Another interesting conclusion is that there is room for improvement in the development of new simulators, particularly for technically demanding procedures.

The results highlight the subjectivity of individual feelings and yet a great homogeneity in the answers given. The survey shows that the participants would like more realism, particularly in the appearance and consistency of the simulator. It is useful to remember that an additional step can easily be inserted between training on simulators and training on live animals. This is the use of animal remains from previous experimental procedures. The use of dead animals increases realism and removes the apprehension associated with handling live animals. It also allows the effectiveness of a given procedure to be easily verified in a simple manner. For example, the subcutaneous administration of a substance can be verified by means of an injection of coloured solutions, with different colours for each participant, and a verification after dissection of the location of the dye. Such verification is difficult to envisage on live animals unless the animal is sacrificed after the exercise has been carried out, which is not compatible with the replacement requirement.

In line with this observation, the authors note that the participants appreciated the observation window available on model B. In some ways this result is not surprising since this window allows for immediate verification of the correct positioning of the gavage tube. In fact, in addition to the search for greater realism, it seems essential to insist on the need to implement on these simulators the possibility of verifying the correct performance of the procedure, whether it is administration or sampling.

Figure 4 shows the large use of ip injections. I do not agree with the authors' proposal (page 13, line 695) that a new simulator should also allow IP injections to be performed. In my opinion, this method of administration only exists because it is essentially a convenience for operators and in no way respects the principle of refinement. Indeed, it is not the recommended route for drugs in veterinary medicine and its use should be discouraged for many substances in research. The recent literature reports numerous failures in both the performance of the procedure and the effectiveness of the administration, which gives rise to great variability because the targeted injection space is not consistently reached.

In the same way, I do not support the idea of optimising the speed of execution of the procedure. If repetition allows to acquire a better dexterity and to reduce errors, the execution time should not be a variable to consider. During the learning phase, it is important to focus on the objective of correct and efficient execution with the least impact on the animal rather than on a rapid execution which adds unnecessary and negative pressure on the operator. In real-life conditions, the demand for speed, if it obeys a logic of pure profitability, is in my opinion incompatible with respect for animal welfare. It is legitimate to think that executing a gesture quickly reduces the stress and constraint on the animal. However, there are other approaches that can reduce stress, in particular a prior and repeated handling of animals or a positive reinforcement. A laboratory animal will be all the more cooperative if it has been previously accustomed to handling, even for gestures considered invasive. This is a key element of the culture of care, for which the only acceptable requirement is that the time needed to obtain the desired results be allocated.

Under these conditions, the ideal of a single, versatile simulator seems unrealistic. It seems more reasonable to me to rely on the complementarity of approaches for a given gesture by means of several simulators of increasing realism so as to allow not only the learning of the gesture itself but also the learning of its impact and consequences for the animal. Following the adage "we learn from our mistakes", an effective simulator must also allow for mistakes to be made and errors to be objectivated. Additionally, an ideal simulator should allow not only the acquisition of new skills but also the maintenance of skills.

On the form of the manuscript, I have few observations or recommendations to make. A few typographical errors need to be corrected:

  • Page 5, Table 4, first column, last line of the table (21) "techniques”
  • Page 8 Figure 1, in the Title the term requirements is misspelled.

Finally, I would like to congratulate the authors on this remarkable work which, in my opinion, should encourage the development of simulators for LAS training.

Reviewer 3 Report

This is an interesting assessment made by participants of the utility of simulators for training potential personal licensees in various techniques in laboratory animal science.  It compares 5 different simulators (for rats) using a questionnaire given after simulator training (Part 1) and after attempting the same techniques on living animals (part 2).  There were no significant differences detected but there were some obvious different responses between participants.  The assessment of competence in the techniques by an independent tutor was not made, only the participants’ feedback on a Likert scale.  Open ended questions helped signal improvements that might be made.  Overall it is well written and understandable but it does need some minor editing for the English by a native speaker.  I would be happy to do that if you send me a Word.doc.

This study is a report of trialling 5 simulators and the participants’ responses to a questionnaire asking them to compare their experiences with the simulators and with using living animals.  I would make the following points.

  1. Lines 702-706: Not clear whether the animal was anaesthetised for all/some/none of the procedures on the live rat, please provide that information in the text and in the appropriate table.Some of the techniques (cardiac puncture, venous bleeds, venous injections, gavage) carried out on living animals would be quite painful and distressing, and it is debateable whether they should be carried out for the purpose of training at this stage.  Normally it would be acceptable training persons using simulators to give them an idea of that is involved.  However, I would recommend training only those persons who will go on to use such techniques and then using an anaesthetised live animal before learning on a potentially painful and distressing technique on a fully conscious living animal.  This is a practical as well as an ethical question and one that very much reflects the overall goal on Refinement.  One of the goals of refinement is not to cause avoidable suffering.  If one trains all persons on live animals it would include those who may only wish to carry out less invasive procedures.

I have many other minor comments which are to do with the English which in general is very good, but may be ambiguous or not the correct context or nuance.

  1. Lines 28 and 41: I suggest: “…may not completely replace…. As stated earlier in the Simple Summary by the word supplementary
  2. Line 50: Russell
  3. Line 54: Suggest “….refine any suffering in the procedures…”
  4. Line 55: laboratory animal science (LAS) not animals
  5. Line 57: implementation, not constitution
  6. Line 59: have to, not are prompted
  7. Lines 61-63: Suggest: “For this purpose, recommendations for education and training were established by experts for the European Commission [7,8] and, in many countries, personal licensees are qualified by species-specific laboratory animal training courses following these suggestions.
  8. Line 76: suturing not suture
  9. Line 79: omit as
  10. Line 88-90: Suggest: However, our previous study was the only one that seems to have dealt with a focussed analysis on the implementation and satisfaction with current simulators for LAS courses.
  11. Line 116/129: National Competent Authorities
  12. Lines 171, 172: omit the word copies.
  13. Lines: 91,103,152,157,255,304,429,589,628, and the tables and questionnaire: Methodical should be replaced by methodological
  14. Table 1: Lancet sizes do not seem to be described correctly. 25G is 0,5mm, and 30G is 0,3mm.  3mm is likely to be a catheter size (e.g. 9FG) not a needle size.  Could it be referring to the length of the lancet. Would 25Gx3mm long be more accurate?
  15. Lines 324-326: There are no data given in Table 4 for this conclusion to be drawn.
  16. Questionnaire Pt2 Q6 - explain ‘haptics in the text: e.g. the perception of objects by touch and proprioception.
  17. It would be helpful to refer to the Questionnaire being analysed in the legend to the Tables
  18. As I have stated above Overall it is well written and understandable but it does need some minor editing for the English by a native speaker. I would be happy to do that if you send me a Word.doc.
